# In Vivo Detection of Metabolic Fluctuations in Real Time Using the NanoBiT Technology Based on PII Signalling Protein Interactions

**DOI:** 10.3390/ijms25063409

**Published:** 2024-03-17

**Authors:** Rokhsareh Rozbeh, Karl Forchhammer

**Affiliations:** Interfaculty Institute of Microbiology and Infection Biology, University Tübingen, Auf der Morgenstelle 28, 72076 Tübingen, Germany; rokhsareh.rozbeh@gmail.com

**Keywords:** protein-fragment complementation assay, NanoLuc, luciferase, 2-oxoglutarate, PII-interacting protein X (PipX), N-Acetylglutamate kinase (NAGK), metabolite sensor

## Abstract

New protein-fragment complementation assays (PCA) have successfully been developed to characterize protein–protein interactions in vitro and in vivo. Notably, the NanoBiT technology, employing fragment complementation of NanoLuc luciferase, stands out for its high sensitivity, wide dynamic range, and straightforward read out. Previously, we explored the in vitro protein interaction dynamics of the PII signalling protein using NanoBiT, revealing significant modulation of luminescence signals generated by the interaction between PII and its receptor protein NAGK by 2-oxoglutarate levels. In the current work, we investigated this technology in vivo, to find out whether recombinantly expressed NanoBiT constructs using the NanoLuc large fragment fused to PII and PII-interaction partners NAGK or PipX-fused to the NanoLuc Small BiT are capable of detecting the metabolic fluctuations in *Escherichia coli*. Therefore, we devised an assay capable of capturing the metabolic responses of *E. coli* cells, demonstrating real-time metabolic perturbation upon nitrogen upshift or depletion treatments. In particular, the PII-NAGK NanoBitT sensor pair reported these changes in a highly sensitive manner.

## 1. Introduction

Protein–protein interactions (PPIs) constitute a complex network within cells, on which almost all biological processes are based, such as DNA replication, transcription, signal transduction, enzymatic reactions, cell-to-cell communication, and membrane transport [1,2]. The three-dimensional conformation and dynamic behaviours of the involved proteins predominantly regulate these interactions [3]. These interactions range from being permanent to transient in nature [1,4].

To investigate protein–protein interactions in living cells, it is desirable to employ biosensor tools that do not disrupt the native cellular context. High sensitivity and high signal to noise read out are important properties for robust and precise analysis. In this context, protein-fragment complementation assays (PCA) have successfully been developed [5]. Among the different types of PCA, the assays based on the small bioluminescence enzyme NanoLuc are of particular interest, due to the high bioluminescence output of this enzyme. Verhoef et al. (2016) developed a NanoLuc based assay where several NanoLuc fragments were generated by cleaving at different loop sites, and a particular pair comprising of the N-terminal 52-amino acid (aa) fragment and the C-terminal 119-aa fragment was selected [6]. Concurrently, Dixon et al. introduced an alternative NanoLuc-based PCA-assay system known as NanoLuc Binary Technology (often referred as split NanoLuc). A cleavage point was strategically positioned between NanoLuc residues 156 and 157, resulting in the generation of two distinct peptide fragments: an 18 kDa large fragment (termed LgBiT) and a small 11-amino acid peptide (SmBiT). To dissect specific protein–protein interactions, the SmBiT peptide underwent engineering to minimize its inherent affinity towards the LgBiT complementation fragment. This strategy ensures that bioluminescence reconstitution, following the fusion of LgBiT and SmBiT fragments to proteins of interest, primarily relies on the interaction of the fusion partner proteins. The PCA is performed by measuring the bioluminescence after addition of suitable substrate, which can be done in vitro or in vivo. In addition to this, a different version of the NanoBiT technology uses a SmBiT fragment with high affinity to LgBiT, termed HiBiT. This system has a different purpose and is used to study protein localization and secretion [7].

In recent years, the NanoBiT technology using the low-affinity SmBiT fragment has been utilized for the precise quantitative analysis of in vitro protein–protein interactions. In mammalian systems, the NanoBiT system was successfully used to analyse the interaction of LgBiT-fused protein phosphatase 2A-B55α holoenzyme of endothelial cells with the SmBiT flotillin-1 protein [8]. To study the SARS-CoV-2 entry mechanism into human epithelial cells, the NanoBiT system was deployed by monitoring the interaction between the receptor-binding domain (RBD) and ACE2 [9]. NanoBiT technology is also increasingly in use for bacterial PPIs. For example, it was employed to verify the interaction between bacterial transcription factors NusB and NusE in *Bacillus subtilis* [10]. Similarly, the interaction RNA polymerase and initiation factor σ in *B. subtilis* was explored using NanoBiT, where the LgBiT was fused to the C-terminus of σA and SmBiT to the N-terminus of the RNA polymerase [10]. The split NanoLuc assay corroborated interactions, including those between the ribosome-associated GTPases (RA-GTPase) Era family and the 16S rRNA endonuclease YbeY, as well as the DEAD-box RNA helicase CshA in *Staphylococcus aureus* [11]. We used the NanoBiT system to accurately characterize and quantify the interaction between the PII signalling protein from the cyanobacterium *Synechocystis* sp. PCC6803 and its interaction partners, PipX and N-acetyl-L glutamate kinase (NAGK).

In addition to in vitro studies, NanoBiT has been extensively utilized for the examination of in vivo protein–protein interactions, mostly in mammalian cells [12,13,14,15]. Within bacterial cells, NanoBiT technology was employed to assess the multimerization ability of HupA, a homologue of the histone-like HU proteins, in *Clostridium difficile*, where HupA was strategically fused to the C-terminus of both SmBiT and LgBiT subunits [16]. Additionally, split NanoLuc assays were utilized to probe interactions between the transmembrane peptidase AgrB and its AgrD propeptide substrate in *Staphylococcus aureus* cells, with AgrB and AgrD being labelled with either N-terminal LgBiT, N-terminal SmBiT, C-terminal LgBiT, or C-terminal SmBiT [17]. In these cases, the primary goal of the NanoBiT technology was to demonstrate the specificity of protein interactions in vivo.

Given the impressive sensitivity and dynamic range of the NanoBiT reporter system, it presents a promising avenue for developing sensors capable of monitoring real-time metabolic changes within living cells through metabolite-dependent protein–protein interactions. The versatile PII signalling protein, with its metabolite-responsive interactions, serves as an ideal platform for such intracellular metabolite reporters. The PII protein has multitasking sensory properties by binding the effector molecules ADP, ATP or Mg-ATP-2-oxoglutarate, and transmitting metabolic information to various receptor proteins. The complex network of PII interactions is comprehensively reviewed in [18]. In cyanobacteria, among the principal interactors are the transcriptional co-activator PipX and the arginine-pathway enzyme N-Acetylglutamate kinase (NAGK) [18,19]. We and others have previously used the Förster resonance energy transfer (FRET) technology to study these interactions, whereby fluorescent proteins were fused to the C-termini of PII and the respective partner proteins [20,21]. Although these attempts led to a better understanding in PII interactions, a major drawback of the FRET system is its very narrow dynamic range and it requires precise adjustment of complex parameters such as the ratio between the partner proteins [22]. Furthermore, it suffers from high signal noise generated by background fluorescence, in particular in photosynthetic cells with high intracellular pigment concentration [23]. The NanoBiT technology avoids these restrictions.

We have previously established PII-based NanoBiT reporters for in vitro analysis of PII complex formation with its receptors PipX and NAGK and were able to precisely quantify association constants in presence of different effector molecule combinations [24]. Maximum bioluminescence signals were achieved when LgBiT was fused to the C-terminal Strep-tag of the PII protein, and SmBiT was fused to the C-terminus of either PipX or NAGK, separated by an appropriately sized flexible linker [24]. Due to the intricate properties of PII signalling proteins [19], the metabolite effects on the formation of PII-NAGK complexes differ from that for PII-PipX complexes. Presence of ATP favours formation of the PII-NAGK complex, whereas 2-oxoglutarate acts antagonistically. This response is quite robust against changes in the ATP/ADP ratio [25]. By contrast, the formation of the PII-PipX complex is favoured by ADP and antagonized by simultaneously binding Mg-ATP-2-oxoglutarate [26]. Subtle changes in the ADP concentrations already modulate the antagonistic effects of 2-oxoglutarate, such that increasing ADP concentrations strongly mitigate the antagonistic effect of 2-oxoglutarate [24,26].

The present work was carried out to find out how the NanoBiT based reporter pairs PII-NAGK or PII-PipX perform in vivo, whether they are able to respond to metabolic perturbations and if they allow in vivo detection of metabolic fluctuations. As a proof-of-concept study, the analysis was carried out in recombinant *Escherichia coli* cells, challenged by nitrogen up- and down-shift treatments, for which the posttranslational [27] and metabolic and response is well established [28].

## 2. Results

### 2.1. Establishment of PII-Based In Vivo NanoBiT Sensors

To co-express the NanoBiT sensor pairs in *Escherichia coli*, the genetic constructs shown in Figure 1 were made. As vector backbone the pACYCDuet-1 plasmid was used, which is designed for equal co-expression of two genes of interest. Briefly, the PII signalling protein was fused at its C-terminus to the large subunit of the split NanoLuc enzyme (LgBiT), connected by an 8 amino acids linker. As interaction partners, either PipX or N-Acetyl-L-Glutamate Kinase (NAGK) were used, which were fused at their C-terminal end to the SmBiT fragment, connected by an 16 amino acids linker as described previously [24]. As a control for maximal luciferase activity, which is independent of complex formation, the PII protein was fused at its C-terminus to full-length NanoLuc (PII-FL). As control for the luminescence background signal, the PII-LgBiT construct lacking a SmBiT partner was used. As a further control to proof the specificity of the PII-based protein interaction, we used a variant of the PII protein with the S49E point mutation, which strongly abrogates PII-NAGK interaction [20,24]. These plasmids were then transformed into *E. coli* BL21 (DE3) cells for bioluminescence measurements and the bioluminescence was measured, as described in the methods section.

The first assays were performed with either the PII-NAGK NanoBiT sensor pair or the PII-FL reporter. Following preliminary trials, we recognized the importance of precisely pre-cultivating the recombinant strains before conducting luminescence measurements, as outlined in the methods section, to ensure reproducible results. Initial experiments for assay development were done with the PII-FL reporter whose signal is independent of protein–protein interaction and only depends on the expression level of the PII-NanoLuc fusion protein. Usually, genes cloned into the pACYCDuet vectors are expressed by inducing the T7 polymerase of the host cells through the addition of IPTG. In the absence of inducer, genes are expressed at a background level. We first measured the relative luminescence signals (RLUs) from cultures, which were induced with 0.5 mM IPTG (OD600 = 0.5–0.7) and of non-induced cultures. The results are summarized in Table 1. After 60 min incubation, the RLUs were recorded. IPTG-induced cells displayed a signal of 6.8 × 10^7^ RLU (STD = 7.2 × 10^6^ RLU), whereas in the absence of IPTG, 4.7 × 10^6^ RLU were recorded, which corresponds to an approximately 15-fold increase by IPTG induction. Likewise, we analysed the RLUs from the PII-NAGK NanoBiT sensor pair under induced and non-induced conditions. Here, we obtained values of 6.0 × 10^6^ or 4.2 × 10^5^ under induced or non-induced conditions, respectively, which corresponds to a 14-fold increase by IPTG induction, a similar fold-increase than with PII-FL. In both cases (induced and non-induced), the signals from the PII-NAGK NanoBiT constructs corresponded to approximately 9% of the signal form the full-length PII-Nanoluc sensor. This indicates the level of reconstitution of the split NanoLuc signal as compared to the signal of full-length NanoLuc enzyme. As a negative control for background noise, luminescence was recorded from cells, which expressed only PII-LgBiT. The background luminescence was in the rage of 3–2 × 10^2^ RLU, which is more than three orders of magnitude below the signal from the non-induced sensor strains. As the signal under non-induced conditions was high enough to allow precise measurement of luminescence, we decided to work under non-inducing conditions, which avoids metabolic burden to the cells. To find out if, under such conditions, the measurement is still sensitive enough to detect even very weak interactions, we used the PII-S49E variant, which previously was shown to bind to NAGK with strongly reduced affinity. In fact, a luminescence signal was detected which was well above the background level and reached approximately 5% of that of the non-mutated PII-NAGK sensor (Table 1). This is in perfect agreement with our previous in vitro PII-NAGK NanoBiT assays, which showed that the binding affinity of PII-S49E to NAGK (in presence of 2 mM ATP) is approximately 5% of that of wild-type PII [24].

These preliminary experiments indicated that the in vivo measurements using the PII-NAGK NanoBiT sensor yielded results compatible with the previous in vitro measurements. Since the affinity of PII-NAGK interaction is strongly modulated by the metabolic state, such that 2-oxoglutarate strongly inhibited complex formation, we next attempted to use the PII-NAGK sensor to monitor changes in metabolic states in real-time upon challenging the cells with different external stimuli.

### 2.2. Monitoring Metabolic Fluctuations Using the PII-NAGK Sensor

To monitor the immediate effects of metabolic perturbation, the recording of bioluminescence should start immediately after onset of the treatment. Previous rapid sampling metabolome analysis in *E. coli* showed that perturbations in nitrogen homeostasis resulted in metabolic responses in the range of seconds to minutes [28]. To establish a rapid assay, we again first used the PII-FL reporter to validate fluctuations in RLU signals. The RLUs were recorded over a period of 10 min in intervals of 10 s (5 s measurement followed by a 5 s delay). The RLU signals decreased over time, and this decrease was apparently caused by oxygen depletion in the measuring cuvette. When the samples were shaken, the luminescence briefly increased but then decreased again rapidly (see Figure 2). After approximately 5–10 min, luminescence decreased to a basal level. In principle, the same dynamics was observed with the PII-NAGK NanoBiT sensor pair. The lack of an initial stable luminescence signal prevented a simple assessment of immediate dynamic responses of the NanoBiT sensor caused by fluctuations in intracellular metabolite pools. However, we could solve this problem by normalizing the RLU response curve of treated samples to the RLU response curve of a reference sample (without perturbation). Normalization was done by dividing the RLUs from every measuring point of the treated sample by the respective RLU of the untreated sample (compare Figure 3A,B).

As a first perturbation experiment and as a proof of the measurement principle, we analysed the response of the PII-NAGK NanoBiT sensor pair towards ammonium upshift treatments. It is well established that ammonium treatment leads to a rapid reduction in 2-oxogulatarate levels [29]. This should lead to intracellular conditions that favour PII-NAGK complex formation. Figure 3A shows the result of this experiment without normalization. In fact, the addition of 40 mM NH_4_Cl led to an immediate RLU increase, but due to the baseline drift described above, luminescence decreases again after 60 s. Nevertheless, the signal remains always higher than the untreated control that was incubated at a constant concentration of 1 mM NH_4_Cl. When ammonium treatment was done with 4 mM NH_4_Cl, this led to less pronounced, but still clearly visible RLU increase over the untreated control. After performing the above-described normalization, it is clearly visible that the addition of 40 mM ammonium caused an immediate increase in the relative signal, which is indicative of increasing PII-NAGK interaction (Figure 3B). After approximately 2 min, a maximum of a two-fold increase was reached, followed by a slower decline and reaching a plateau that was approximately 50% over the level of the untreated sample. Treatment with 4 mM NH_4_Cl showed initially a very similar dynamics, reaching a maximum after 2 min, but the signal only increased by 50%, and then rapidly returned to the initial level of the untreated sample. To reveal if this response really reflected changes in the state of the split-NanoLuc PII-NAGK complex and was not caused by indirect metabolic effects of ammonium treatment on the luminescence reaction itself, the same assay was performed with the PII-FL reporter. As shown in Figure 3C,D, the same ammonium treatment as above had almost no influence on the normalized RLU curve. This demonstrates that the relative signal increase shown by the PII-NAGK sensor pair is indeed caused by increased PII-NAGK NanoBiT assembly, indicative of dropping 2-oxoglutarate levels.

Next, the response of the PII-NAGK NanoBiT sensor pair to inhibition of nitrogen assimilation was tested. Therefore, cells were treated with either 0.1 mM or 1 mM L-methionine-sulfoximine (MSX), a specific and potent inhibitor of glutamine synthetase, or as a reference, without MSX treatment. The result is shown in Figure 4A,B: Addition of 0.1 or 1 mM MSX caused a rapid decrease in PII-NAGK NanoBiT complex formation, visible as decrease in RLU. After 6 min, the sensor signal in presence of 0.1 mM MSX or 1 mM MSX decreased to approximately 39% or 27%, respectively, of the untreated control. Again, the control experiment with the PII-FL sensor showed only a very minor response (Figure 4C). Finally, the response towards nitrogen starvation was investigated. Immediately after nitrogen step-down, the signal from the PII-NAGK NanoBiT sensor pair decreased until it reached a basal level, which corresponded to 35% of the untreated control after approximately 5–6 min and then stayed constant (Figure 4D,E). Again, the PII-FL reported did not respond to nitrogen downshift (Figure 4F), revealing that the observed signal decrease of the split-NanoLuc sensor pair represents dissociation of the PII-NAGK complex, which is expected from increasing levels of 2-oxoglutarate caused by nitrogen deprivation. Altogether, the above experiments showed that the PII-NAGK NanoBiT sensor pair was able to report in real time the dynamics of metabolic fluctuations caused by perturbing nitrogen homeostasis.

### 2.3. Monitoring Metabolic Fluctuations Using the PII-PipX Sensor

Having confirmed the response of the PII-NAGK sensor pair to nitrogen assimilation perturbations, we subsequently employed the PII-PipX sensor pair in analogous experiments (Figure 5A–F), following the same experimental protocol as described above.

In this case, the 40 mM NH_4_Cl treatment caused a more rapid initial signal increase as compared to the PII-NAGK sensor and the elevated RLU remained at high level. The effect of 4 mM was delayed and reached a level between the untreated control and the 40 mM NH_4_Cl treatment, similar to that which we observed for the PII-NAGK sensor pair. A quite different result was obtained in the nitrogen step-down experiments. Here, the PII-PipX sensor showed almost no response to the inhibition of nitrogen assimilation by MSX treatment. In the nitrogen step-down experiment, an immediate decrease in RLUs was observed. However, this decrease was followed by a subsequent increase to nearly the initial RLU ratio. This clearly demonstrates that the PII-PipX NanoBiT sensor detects metabolic perturbations in a distinct manner compared to the PII-NAGK NanoBiT sensor. This agrees with the corresponding in vitro analysis of these sensor pairs, which showed that the PII-NAGK complex primarily responds to fluctuations in the 2-oxoglutarate levels, whereas PII-PipX interaction is strongly modulated by the presence of ADP.

### 2.4. Monitoring Metabolic Response upon Complete Nutrient Deprivation

The results obtained above suggested that in vivo fluctuations in 2-oxoglutarate levels are most sensitively detected by the PII-NAGK NanoBiT sensor. Therefore, we ultimately employed this sensor to monitor the metabolic response of *E. coli* cells towards an extremely stressful condition: shifting exponentially growing cells, cultivated in M9 medium, into distilled water: under these conditions, all nutrients are simultaneously removed.

We were curious to see the luminescence output from the PII-NAGK NanoBiT sensor. The experiment was carried out as described above and the PII-FL reporter was used as a control to monitor global effects on luminescence, which are independent on Split NanoBiT complex formation. As shown in Figure 6, the normalized signal measured with the PII-NAGK NanoBiT sensor increased immediately after the transfer and reached a 10-fold higher level 5 min after the transfer. By contrast, in the PII-FL control, only a marginal slow constant increase in relative RLU signal was visible. The strong increase detected in the PII-NAGK NanoBiT sensor cells indicates a dramatic reduction in the 2-oxoglutarate levels in response to shifting the cells to distilled water. This suggests that the cells consume the remaining TCA cycle metabolites, once nutrients are completely deprived.

## 3. Discussion

This work showed successful application of NanoBiT reporter constructs to follow in real-time changes in cellular 2-oxoglutarate levels. The in vivo interaction of recombinantly expressed PII-NAGK NanoBiT or PII-PipX NanoBiT sensor pairs in *E. coli* results in the reconstitution of the split NanoLuc reporter, which results in luminescence signals after adding the luminescence substrate. The background expression of the gene fusions form the pACYC-Duet vector, without addition of the inducer IPTG, was high enough to accurately measure the specific NanoBiT signals. Even low affinity interactions, such as the binding of the low-affinity PII-S49E variant to NAGK could be detected. Remarkably, a similar degree of NanoBiT reconstitution as obtained previously with purified proteins [24], demonstrating the reliability of the results.

Comparing the results with the two different PII interaction partners showed that the response of the NanoBiT sensors towards ammonium upshift yielded similar results for both, the PII-NAGK NanoBiT and PII-PipX NanoBiT senor pair. Previous metabolome analysis carried out by rapid sampling of *E. coli* cells following the addition of 10 mM NH_4_Cl to nitrogen-limited cells showed a fast decline of 2-oxoglutarate levels and concomitantly to a rapid increase of glutamine levels [28]. Since the cyanobacterial PII and NAGK proteins do not respond to glutamine [19], it can be safely assumed that the observed increase in NanoBiT signals is indeed driven by the dropping 2-oxoglutarate levels. As also revealed by direct metabolite measurement, these changes occur in the range of a few minutes. After 2–3 min, the response is at its maximum [27,28].

In contrast to ammonium upshift treatments, only the PII-NAGK NanoBiT sensor was able to detect metabolic fluctuations induced by either the inhibition of nitrogen assimilation through glutamine synthetase inhibition or the removal of combined nitrogen sources. Previous metabolome analysis with *E. coli* cells revealed a rapid increase in 2-oxoglutarate levels, which would cause dissociation of PII-NAGK complexes. This agrees well with the observed rapid decline in the normalized RLU signals, which indicates dissociation of the PII-NAGK complex in vivo. Inhibition of glutamine synthetase or removal of combined nitrogen (ammonium) resulted in a similar response by the PII-NAGK NanoBiT sensor. This implies that transferring the *E. coli* cells into ammonium-free medium results in an immediate arrest of glutamine synthetase activity, like addition of the inhibitor MSX. In contrast to the PII-NAGK NanoBiT sensor, the PII-PipX NanoBiT sensor did not show a clear RLU decline. Previous in vitro analyses of PII interactions with NAGK or PipX work showed that in contrast to PII-NAGK complexes, the PII-PipX interaction is very sensitive towards subtle changes in ADP levels [25,26]. As the ADP concentrations increase, the affinity between PII and PipX increases. As shown by Zeth et al. [26], the conformation of PII in the ADP complex matches the conformation of PII in complex with PipX [26] and therefore, PipX and ADP act synergistically in binding to PII. The lack of efficient dissociation of the PII-PipX complex upon inhibition of glutamine synthetase reaction—in contrast to dissociation of the PII-NAGK NanoBiT pair imposed by increasing 2-oxoglutarate levels could be caused by subtly increasing ADP concentrations, which would prevent dissociation of PII-PipX complex but not that of the PII-NAGK complex. Dissociation of the latter complex was shown to be insensitive towards subtle fluctuations of ADP levels [26].

Overall, the PII-NAGK NanoBiT sensor described here appears to be a useful reporter to detect rapid fluctuations of cellular 2-oxoglutarate levels, both for increasing or decreasing levels. In our final experiment, we used this sensor to monitor the cellular response upon transferring exponentially growing cells into completely nutrient-free water. The reporter showed a maximal signal increase indicating full association of PII and NAGK, indicating extremely low 2-oxoglutarate levels. This metabolic response appears reasonable: in the absence of nutrients, the metabolites of the TCA cycle would be completely consumed for respiration until reaching a minimum. In a similar way, this reporter construct, cloned into appropriate expression plasmids, could provide easy and fast detection of real-time metabolic responses of bacterial cells towards various conditions. Eventually, when cloned into appropriate expression plasmids, the sensor may also be applied for metabolite sensing in other cellular systems, like eukaryotic cell cultures, as NanoBiT reporters have extensively been used in mammalian cells [12,13,14,15].

## 4. Materials and Methods

Cloning of NanoBiT constructs: to create the split NanoLuc constructs for in vivo study, we used as vector backbone the pACYCDuet-1 plasmid (Addgene, Watertown, MA, USA), which is designed for the coexpression of two target genes [30]. Therefore, the vector contains two multiple cloning sites, into which the two NanoBiT partner constructs were cloned. Each cloning site is preceded by a T7 promoter, lac operator and ribosome binding site. The same NanoBiT gene fusions were used as previously described for in vitro investigation of PII interactions with its target proteins [24]. The gene encoding PII-LgBiT was cloned in the linearized vector with *Nco*I (New England Biolabs, Frankfurt am Main, Germany) in the first cloning site. NAGK-SmBiT or PipX-SmBiT were cloned into the *Nde*I (New England Biolabs, Frankfurt am Main, Germany) second cloning site for pACYCDuet vector following the protocol described by Gibson cloning previously [31]. The plasmid DNA was transformed into *E. coli* DH10β, and cultivated at 37 °C overnight on agar plates using the chloramphenicol antibiotic with the final concentration of 25 mg/mL for selection. Positive colonies were detected using colony PCR (Labcycler Basic, SensoQuest, Göttingen, Germany). These were then moved to Lysogeny broth (LB) [32] enriched with the chloramphenicol antibiotic and allowed to grow overnight at 37 °C. The plasmid DNA was extracted from the cells utilizing the Monarch Plasmid Miniprep Kit (New England Biolabs, Frankfurt am Main, Germany) following the guidelines provided by the manufacturer. The DNA sequences were confirmed by the GATC LIGHTRUN sequencing services (Eurofins Genomics, Ebersberg, Germany). After verification, the plasmids were transformed into electrocompetent *E. coli* BL21 (DE3) cells [33].

### Cultivation of NanoBiT Expressing Cells and Luminescence Measurement

Cells from LB agar plates with the appropriate NanoBiT constructs were first cultivated overnight in LB with chloramphenicol antibiotic, from where an aliquot was removed to inoculate cells in M9 mineral salts medium supplemented with 0.4% glucose to an initial optical density (OD600) of 0.07. Cells were allowed to grow at 37 °C until an OD600 of 0.9 was reached, from where a second transfer of cells into M9 medium occurred, yielding a starting OD600 of 0.1. Cultivation was continued until an OD600 between 0.5 to 0.7 was reached. At this point, 500 µL aliquots of the culture were removed and used for bioluminescence recording, immediately after adding the bioluminescence reagent (Promega, Walldorf, Germany) together with the specific treatment that should result in metabolic perturbation. To assess the effects of various substances and inhibitors on live *E. coli* cells using real-time bioluminescence measurement, 20 µL of the selected concentration of the component was combined with 10 µL of Nano-Glo^®^ Live Cell Reagent (made by mixing 1 part of Nano-Glo^®^ Live Cell Substrate with 19 parts of Nano-Glo^®^ LCS Dilution Buffer) (Promega, Walldorf, Germany) in a luminescence reaction tube. Following this, 500 µL of the bacterial culture was introduced to this mixture. The resulting luminescence was then measured for 5 s. with 5 s. delay using a Sirius Luminometer (from Berthold Detection System, Bad Wildbad, Germany) operated by FB12 Sirius Software, version 3.2 (Berthold Technologies, Bad Wildbad, Germany)for the indicated length of time. Measurement was immediately started by closing the luminometer’s door. For each tested condition, at least three independent measurements (biological replicates) were carried out, from which the average RLU and the standard deviation at each time point were calculated. The graphs show the average value at each measured time point. For clarity of the presentation, the error bars are not shown. However, the average values of the standard deviations at any time point were calculated and the value is given in the legends to the figures.

For nitrogen depletion experiments, cells were cultivated as described above in M9 media until they reached an OD600 of approximately 0.6. Then, a 500 µL aliquot was harvested by centrifugation for 60 s at 10,000 rpm (Thermo Scientific Pico 17 Microcentrifuge, Sindelfingen, Germany). Following the removal of the supernatant, the pellet was resuspended in 500 µL of M9 media without a combined-nitrogen source, and luminescence was immediately measured as described above.

## 5. Conclusions

We report here how the metabolite-sensitive protein interactions of the PII signalling protein were used to develop a sensor for real-time monitoring of metabolic fluctuations within living *Escherichia coli* cells. The PII-interactions reconstitute bioluminescence, sensitive towards metabolic perturbations, from in vivo expressed NanoBiT reporter constructs. We established an assay at negligible background luminescence, where the signals exhibit exceptional sensitivity and dynamic capabilities. Interaction of PII with its receptor NAGK in particular offers a robust platform for the detection of intercellular 2-oxoglutarate fluctuations. We could monitor the rapid metabolic responses of *E. coli* with a resolution of ten seconds in response to changing nitrogen supply. This study lays the groundwork for broader applications in metabolic studies, which should be possible in all organisms, in which the NanoBiT constructs can be expressed in vivo.

## Figures and Tables

**Figure 1 ijms-25-03409-f001:**
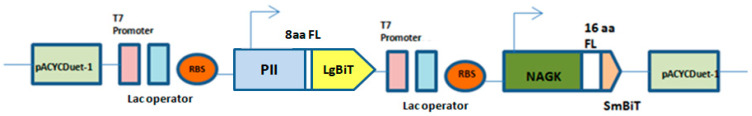
Expression vectors and fusion protein constructs. Schematic overview of PII-LgBiT—NAGK-SmBiT, PII-LgBiT—PipX-SmBiT, PII-Full length (FL), PII (S49E)-LgBiT—NAGK-SmBiT and PII-LgBiT proteins. Flexible linker with 8 amino acids (8aa) fused to LgBiT and flexible linker with 16 amino acids (16aa) fused to SmBiT in constructs.

**Figure 2 ijms-25-03409-f002:**
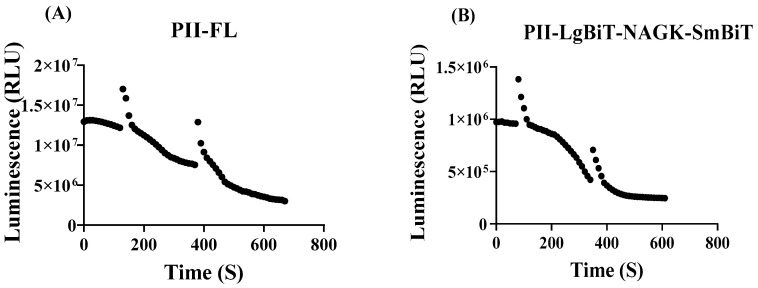
Sensitivity of luminescence intensity (in RLU) in the experimental set-up towards oxygen consumption during the time-course experiments. (**A**) Control measurement using the PII-FL reporter. Measurement was briefly interrupted at time 120 s and 370 s and the samples were shaken and directly measured again. (**B**) As in part (**A**) but using the PII-LgBiT—NAGK-SmBiT sensor; here, the cells were shaken at 80 s and 320 s.

**Figure 3 ijms-25-03409-f003:**
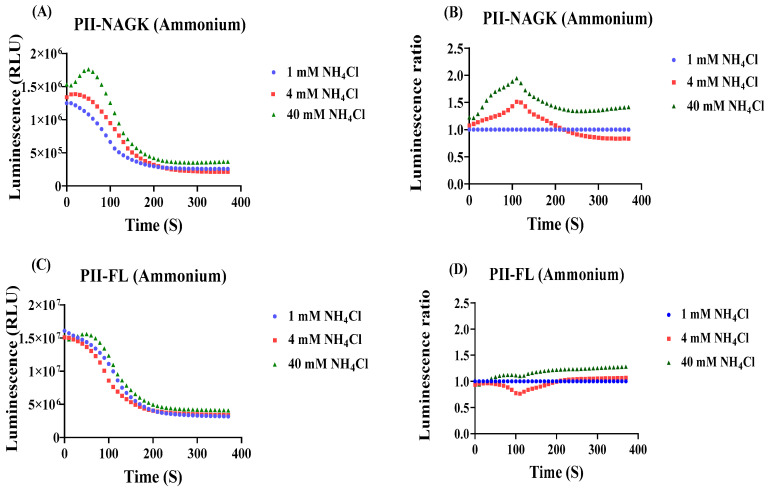
Luminescence response of the PII-NAGK NanoBiT sensor towards ammonium upshift treatments (**A**,**B**) and as control, of the PII-FL reporter under identical test conditions (**C**,**D**). (**A**) Time course of the luminescence signal (RLU) after addition of luminescence reagent in untreated sample (1 mM NH_4_Cl), and to samples, where the NH_4_Cl concentration was increased to 4 mM or 40 mM. (**B**) Normalization of the RLU response curve to the RLU response curve of the reference sample (untreated, 1 mM NH_4_Cl) (**C**,**D**): as part (**A**,**B**) but using the PII-FL reporter for comparison. Average values from three measurements are shown and error bars are removed for a better comparison. The average standard deviation (STD) of each of the three replicates of the PII-NAGK sensor for ammonium treatments was less than 23%. The average standard deviation (STD) of each of the three replicates of the PII-FL sensor for ammonium treatments was less than 14%.

**Figure 4 ijms-25-03409-f004:**
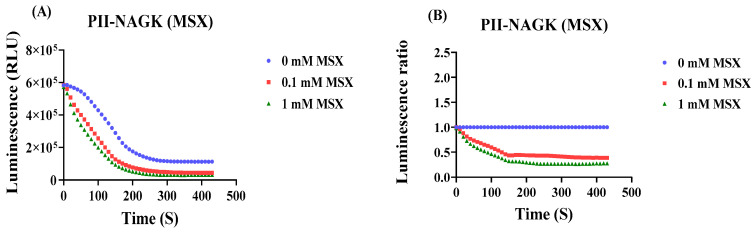
Luminescence response of cells carrying the PII-NAGK NanoBiT sensor following inhibition of ammonium assimilation by methionine sulfoximine (MSX) or following nitrogen starvation. For comparison, the normalized RLU curves of the PII-FL reporter subjected to the same treatments are shown (**E**,**F**). (**A**) RLU time-course measurement of the PII-NAGK NanoBiT sensor in the absence and presence of 0.1 mM and 1 mM MSX. (**B**) Normalizing the RLU response curve to the untreated reference (0 MSX). (**C**) Normalized RLU response curve upon MSX treatment using the PII-FL reporter. (**D**) RLU time-course measurement following nitrogen depletion. (**E**) Normalizing the RLU response curves shown in (**D**) to the untreated reference. (**F**) Normalized RLU response curve upon nitrogen-deprivation using the PII-FL reporter. Average values from three measurements are shown and error bars are removed for a better comparison. The average standard deviation (STD) of each of the three replicates of the PII-NAGK sensor for inhibition of ammonium assimilation by MSX and nitrogen starvation was less than 25%. The average standard deviation (STD) of each of the three replicates of the PII-FL sensor for inhibition of ammonium assimilation by MSX was less than 15%. The average standard deviation (STD) of each of the three replicates of the PII-FL sensor for nitrogen starvation was less than 13%.

**Figure 5 ijms-25-03409-f005:**
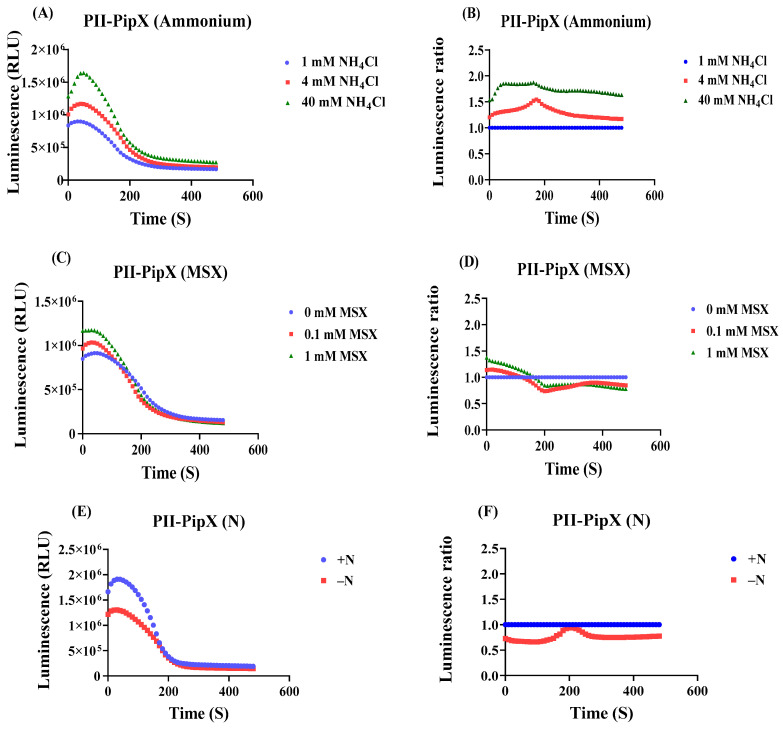
Effect of ammonium addition, MSX treatment and nitrogen starvation on luminescence after addition of luminescence reagent to cells carrying the PII-LgBiT—PipX-SmBiT sensor construct. (**A**) Time course of the luminescence signal (RLU) in untreated sample (1 mM NH_4_Cl), and to samples, where the NH_4_Cl concentration was increased to 4 mM or 40 mM. (**B**) Normalizing the RLU response curve of PII-PipX sensor to the reference sample. (**C**) RLU time course of PII-PipX NanoBiT sensor in the absence and presence of 0.1 mM and 1 mM MSX. (**D**) RLU response curves of (**C**) normalized to the reference sample. (**E**) RLU time course of PII-PipX NanoBiT sensor following nitrogen depletion. (**F**) RLU response curves of (**E**) normalized to the reference sample. Average values from three measurements are shown and error bars are removed for a better comparison. The average standard deviation (STD) of each of the three replicates of the PII-PipX sensor for ammonium treatments was less than 21%. The average standard deviation (STD) of each of the three replicates of the PII-PipX sensor for inhibition of ammonium assimilation by MSX was less than 24%. The average standard deviation (STD) of each of the three replicates of the PII-PipX sensor for nitrogen starvation was less than 23%.

**Figure 6 ijms-25-03409-f006:**
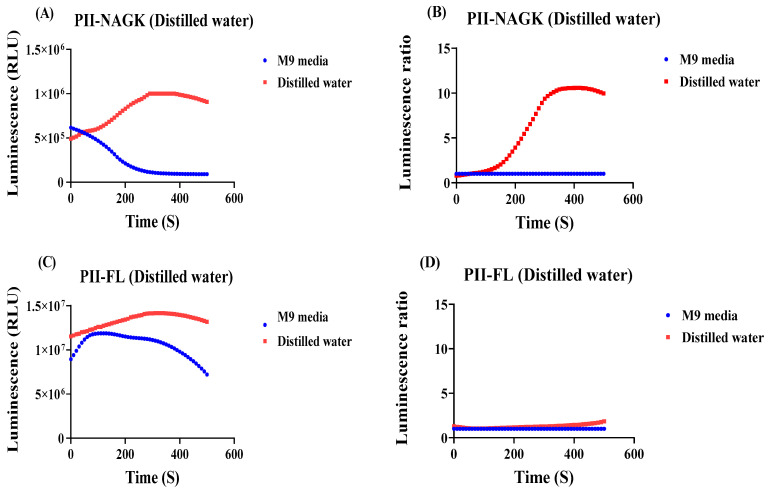
Detection of the metabolic response of E. coli cells shifted from M9 medium into distilled water, using the PII-LgBiT—NAGK-SmBiT sensor or as a control the PII-FL-reporter. (**A**) RLU response curve from the PII-LgBiT—NAGK-SmBiT sensor of treated sample (red) and untreated reference sample (blue) (**B**) RLU response curves of (**A**) normalized to the reference sample. (**C**) RLU response curve from the PII-FL reporter of treated sample (red) and untreated reference sample (blue). (**D**) RLU response curves of (**C**) normalized to the reference sample. Average values from three measurements are shown and error bars are removed for a better comparison. The average standard deviation (STD) of each of the three replicates of the PII-NAGK sensor for nutrient deprivation in distilled water was less than 18%. The average standard deviation (STD) of each of the three replicates of the PII-FL sensor for nutrient deprivation in distilled water was less than 11%.

**Table 1 ijms-25-03409-t001:** Comparison of NanoBiT signals from M9 grown reporter strains with and without IPTG induction

	(−IPTG)		(+IPTG)	
Constructs	RLU Signal	%	RLU Signal	%
PII-FL	4,721,437 ± 623,462	100	68,198,243 ± 7,174,184	100
PII-LgBiT-NAGK-SmBiT	422,034 ± 81,375	9	6,072,362 ± 545,005	9
PII(S49E)-LgBiT-NAGK-SmBiT	22,902 ± 2685	0.5	nm	nm
PII-LgBiT	254 ± 42	0.005	nm	nm

Luminescence signals (in RLU) derived from the PII-NanoLuc Full length (PII-FL) reporter, the PII-LgBiT-NAGK-SmBiT sensor pair, the PII(S49E)-LgBiT-NAGK-SmBiT sensor pair and the PII-LgBiT reporter in absence and presence of inducer IPTG after 60 min of induction. nm = not measured. Average and standard deviation of three replicates is shown.

## Data Availability

Data are contained within the article.

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
