# Peer review of "In Vivo Detection of Metabolic Fluctuations in Real Time Using the NanoBiT Technology Based on PII Signalling Protein Interactions"

_ijms, 2024, doi:10.3390/ijms25063409_

Round 1

Reviewer 1 Report

Comments and Suggestions for Authors

Rokhsareh Rozbeh and Karl Forchhammer combined interaction between PII and its receptor protein NAGK regulated by 2-oxoglutarate and NanoBiT to detect changes in 2-oxoglutarate concentrations in E. coli in real time. This is a very interesting work. Using protein interactions to detect changes in metabolism is not only practical, but also very enlightening.

I have only a few suggestions as follows:

1. How should the academic community obtain the plasmids mentioned in the manuscript? Will these plasmids be shared on addgene?

2. The diagrams in Figure 3~6 need to be combined into one big picture.

3. The results section contains too much method detail and the methods section is too short. The author needs to rewrite both sections.

4. Can this method migrate to mammalian cells? I hope to add something about this in the discussion section.

Comments on the Quality of English Language

To ensure that your work meets the highest standards of scientific communication and readability, I strongly recommend engaging a professional third-party service for both scientific and language polishing.

Author Response

Reviewer 1. 

  1. How should the academic community obtain the plasmids mentioned in the manuscript? Will these plasmids be shared on addgene?

We are open to distributing the plasmids to any interested academic community for their research endeavors. Like in the past, we have delivered our FRET based PII sensor construct to a large audience. The same will hold true to the constructs described here.

  1. The diagrams in Figure 3~6 need to be combined into one big picture.

Thank you for your suggestion. While we understand the desire for a more consolidated visual representation, combining all four figures into one big picture might lead to complexity and potential confusion due to the diverse content. We have arranged the graphs side by side in each figure to enhance clarity and facilitate a more coherent visual representation in the revised manuscript.

  1. The results section contains too much method detail and the methods section is too short. The author needs to rewrite both sections.

This comment mostly apply for the section 2.1. (in particular, the assay development). Although assay development is usually part of the methods, in this particular case, it is integral part of the results, since this is a proof-of concept study. Therefore, the assay itself is an important result and needs to be detailed in the results section. We therefore decided to leave the description of assay development, together with the corresponding Table 1, in the results section. Nevertheless, we have changed some more technical details to the methods section, as suggested by the reviewer.

  1. Can this method migrate to mammalian cells? I hope to add something about this in the discussion section.

Thank you for your inquiry. As discussed in detail in the introduction section, the NanoBiT technology has been successfully applied to mammalian cells. Considering the comprehensive coverage in the introduction, we have intentionally refrained from reiterating this point in the discussion section to avoid redundancy. We believe that the introduction provides a robust foundation for the method's application to mammalian cells. However, to emphasize this important point, we mention the potential application again in the discussion and the new “Conclusion” section.

Reviewer 2 Report

Comments and Suggestions for Authors

During the work, a NanoBiT-type luciferase reporter system was obtained and successfully used to characterize the metabolic perturbations of 2-oxoglutarate in E coli cells exposed to increased and depleted nitrogen. For this complementation assay, reporter pairs of the PII signaling protein with its partner proteins NAGK and PipX fused to luciferase fragments were selected.

The experiments performed are described in detail and clearly; the results confirm that NanoBiT based reporter pairs PII-NAGK or PII-PipX can be used to monitor real-time changes in cellular 2-oxoglutarate levels. In addition, the authors compared the performance of the two sensors and offered an explanation for why the PII-PipX pair did not work in some cases.

In general, the authors clearly demonstrated that similar reporter systems can be used for easy and fast detection of real-time metabolic responses towards various conditions in bacterial cells.

At the same time, I would like to note that the work would have benefited significantly if there had been a PII-regulatory network diagram, including its allosteric regulators and protein partners.

Minor.

In fig. 3, it’s probably worth putting panel E after B, which corresponds to the order in which the experiment is described in the text.

Line 442 – BL21(DE3)

Comments on the Quality of English Language

Minor editing of English language required

Author Response

Reviewer 2.

During the work, a NanoBiT-type luciferase reporter system was obtained and successfully used to characterize the metabolic perturbations of 2-oxoglutarate in E coli cells exposed to increased and depleted nitrogen. For this complementation assay, reporter pairs of the PII signaling protein with its partner proteins NAGK and PipX fused to luciferase fragments were selected.

The experiments performed are described in detail and clearly; the results confirm that NanoBiT based reporter pairs PII-NAGK or PII-PipX can be used to monitor real-time changes in cellular 2-oxoglutarate levels. In addition, the authors compared the performance of the two sensors and offered an explanation for why the PII-PipX pair did not work in some cases.

In general, the authors clearly demonstrated that similar reporter systems can be used for easy and fast detection of real-time metabolic responses towards various conditions in bacterial cells.

At the same time, I would like to note that the work would have benefited significantly if there had been a PII-regulatory network diagram, including its allosteric regulators and protein partners.

We appreciate your suggestion. Considering the extensive range of interaction partners for the PII protein, each with unique interacting mechanisms, incorporating a detailed network diagram might introduce a substantial amount of information that goes beyond the current scope of our study. However, for more clarity, we explicitly refer now to a recent review article, where these PII interactions have been comprehensively been described (Ref 18), see page 2.

Minor.

In fig. 3, it’s probably worth putting panel E after B, which corresponds to the order in which the experiment is described in the text.

Thank you for your thoughtful suggestion regarding the arrangement of panels in figures. We would like to clarify that there is no Panel E in Figure 3, probably you ment fig. 4. Therefore, we reordered the panels in Fig. 4 so that Panel E follows Panel B, aligning with the sequence described in the text. We appreciate your careful review and hope this clarification addresses your concern

Line 442 – BL21(DE3)

Thank you for bringing attention to Line 442. DE3 was added to the text.

Reviewer 3 Report

Comments and Suggestions for Authors

1-    The keywords which is present also in the title should be replaced with other explaining the work.

2-    Authors should follow the Journal format i.e. materials and method followed by results and discussion.

3-    The comments on figure 4 (A-F) should be present in the same page after figure to easily followed up by reader so the figures size should be reduced and be a side to side in one page followed by the comment in the same page.

4-    In line 417: I a similar way, I think it should be replaced with in a similar way.

5-    Conclusion is missing.

Author Response

Reviewer 3.

1-    The keywords which is present also in the title should be replaced with other explaining the work.

We acknowledge your suggestion to replace keywords present in the title with more explanatory terms. In our revised manuscript, we replaced the keyword “PII interacting protein” by the names of PII interacting proteins (PipX and NAGK)

2-    Authors should follow the Journal format i.e. materials and method followed by results and discussion.

Thank you for your valuable feedback. We followed the Journal format by arranging the materials and methods section before presenting the results and discussion in our revised manuscript.

3-    The comments on figure 4 (A-F) should be present in the same page after figure to easily followed up by reader so the figures size should be reduced and be a side to side in one page followed by the comment in the same page.

Thank you very much for this effective comment. In these cases, all the modifications for the figures have been done.

4-    In line 417: I a similar way, I think it should be replaced with in a similar way.

Thank you for pointing out the oversight in line 417. The phrase 'I a similar way' has been revised to 'in a similar way' in our manuscript.

5-    Conclusion is missing.

Thank you for bringing this to our attention. We included the comprehensive conclusion in our manuscript.

Reviewer 4 Report

Comments and Suggestions for Authors

The manuscript represents the research conducted by Rokhsareh Rozbeh et al employs NanoBiT technology to explore protein-protein interactions in vivo, using an interaction of PII-NAGK and PII-PipX proteins in Escherichia coli. The results presented in the manuscript show that the luminescence signals from NanoBiT reconstitution detect metabolic fluctuations. The manuscript is well-written, and the results presented supports the claim made by authors. I have following suggestions for the author to improve this manuscript:

1.      A brief insights into NanoBiT assay limitations regarding experimental conditions like cell density or growth phase will be helpful to get reproduceable results.

2.      The discussion on possible causes of background luminescence and approach applied to characterize them will be helpful to interpret NanoBiT signals effectively.

3.      Complimenting NanoBiT data with metabolomics or transcriptomics would strengthen the conclusions and provide additional validation.

Author Response

 Reviewer 4.

The manuscript represents the research conducted by Rokhsareh Rozbeh et al employs NanoBiT technology to explore protein-protein interactions in vivo, using an interaction of PII-NAGK and PII-PipX proteins in Escherichia coli. The results presented in the manuscript show that the luminescence signals from NanoBiT reconstitution detect metabolic fluctuations. The manuscript is well-written, and the results presented supports the claim made by authors. I have following suggestions for the author to improve this manuscript:

  1. A brief insights into NanoBiT assay limitations regarding experimental conditions like cell density or growth phase will be helpful to get reproduceable results.

Thank you for your valuable feedback. We appreciate your suggestion to provide insights into the experimental challenges related to the NanoBiT assay. In Figure 2, we address the challenge concerning the system's sensitivity to oxygen. To provide further clarity, we conducted time point experiments for each treatment. Initially, we observed that pipetting samples could alter the soluble oxygen levels, impacting luminescence signals. Consequently, we modified our approach to include time course luminescence measurement, ensuring a more controlled and reproducible experimental setup.

  1. The discussion on possible causes of background luminescence and approach applied to characterize them will be helpful to interpret NanoBiT signals effectively.

As shown in the “assay development” section, the background luminescence is extremely low, usually below 0,.1 % of the specific signal and therefore negligible. We highlight the advantage of Bioluminescence  as compared to fluorescence methods in terms of sensitivity and low background noise in several places (introduction, discussion, conclusion)

  1. Complimenting NanoBiT data with metabolomics or transcriptomics would strengthen the conclusions and provide additional validation.

We fully agree that it is important to compare the responses  of the NanoBiT reporters with directly measured parameters at the metabolite and protein level. Therefore, we used metabolic perturbation experiments (nitrogen upshift and downshift) for which extensive data on the metabolic consequences and the response at the protein level are available. The respective studies have been cited. In this regard we added one more reference, where the posttranslational modification of proteins in response to these N-shift experiments was studied (new Ref 27), revealing a similar timing of responses, albeit at lower resolution that the here reported luminescence monitoring.

Further changes:

We added a short acknowledgment to mention a visiting student that for a short period of time took part in initial assay development experiments.